# Photochemistry of Metal Nitroprussides: State-of-the-Art and Perspectives

Paula M. Crespo [1], Oscar F. Odio [2],* and Edilso Reguera [1],*

[1] Instituto Politécnico Nacional, Centro de Investigación en Ciencia Aplicada y Tecnología Avanzada, U. Legaria, Ciudad de México 11500, Mexico; moncbarrera@gmail.com
[2] CONACyT-Instituto Politécnico Nacional, Centro de Investigación en Ciencia Aplicada y Tecnología Avanzada, U. Legaria, Ciudad de México 11500, Mexico
* Correspondence: odiochacon@gmail.com (O.F.O.); edilso.reguera@gmail.com (E.R.)

**Abstract:** This contribution summarizes the current state in the photochemistry of metal nitroprussides, which is dominated by the electronic structure of the nitrosyl group. From the combination of p orbitals of the nitrogen and oxygen atoms in the $NO^+$ ligand, a $\pi^*NO$ molecular orbital of relatively low energy is formed, which has $\pi^*_{2px}$ and $\pi^*_{2py}$ character. This is a double degenerate orbital. When the nitrosyl group is found coordinated to the iron atom in the nitroprusside ion, the availability of that low energy $\pi^*NO$ orbital results in light-induced electronic transitions from the iron atom $d_{xy}$, $d_{xz}$ and $d_{yz}$ orbitals, $2b_2$ (xy) $\to$ 7e ($\pi^*NO$) and 6e (xz,yz) $\to$ 7e ($\pi^*NO$), which are observed at 498 and 394 nm, respectively. These light-induced transitions and the possibility of NO isomer formation dominate the photochemistry of metal nitroprussides. In this feature paper, we discuss the implications of such transitions in the stability of coordination compounds based on the nitroprusside ion in the presence of water molecules for both 3D and 2D structures, including the involved degradation mechanisms. These photo-induced electronic transitions modify the physical and functional properties of solids where the nitroprusside ion forms part of their structure and appear as an opportunity for tuning their magnetic, electrical, optical and as energy-applied materials, for instance. This contribution illustrates these opportunities with results from some recently reported studies, and possible research subjects, even some not explored, are mentioned.

**Keywords:** photochemistry of metal nitroprussides; stability of transition metal nitroprussides; meta-stable states in the nitroprusside ion; photo-isomerization; photochemistry

## 1. Introduction

Metal nitroprussides are salts of the pentacyanonitrosylferrate(II) ion, $[Fe(CN)_5NO]^{2-}$, which were first reported in 1849 by Playfair [1]. Of these six ligands, only the five CNs at their N ends are available to bond a metal center (T); the $NO^+$ moiety remains unlinked at its O end [2,3]. Such salts can be formed with alkaline, alkaline earth [4], transition [5–7] and even rare-earth metals [8] and can even form 2D structures when the axial CN remains unlinked at its N end, resulting in the formation of layered coordination polymers [9]. Certain organic ligands can disrupt the $CN_{Ax}$-T bond to occupy the metal axial coordination sites [10–13].

These organic molecules behave as a pillar between adjacent layers. Since the metal nitroprussides discovery, many studies related to their physical, chemical and electronic properties have been performed and were recently reviewed by Reguera et al. [14]. One of the most relevant properties is related to the hypotensive action of the sodium nitroprusside, NaNP, which was demonstrated in 1929 and made the NaNP useful in cases of severe hypertension [15]. This feature is due to the presence and release of the NO moiety when the NaNP is dispersed in an aqueous solution and receives illumination.

Metal nitroprussides can absorb light through three mechanisms [16]: (1) metal-ligand-charge transfer (MLCT); (2) band-to-band transitions; (3) d-d transitions when an

appropriate transition metal (T) is involved. The photochemistry of metal nitroprussides is dominated by the first mechanism, related to the photoactive properties of the nitrosyl group. It is known that the MLCT is the main process in the photochemistry of transition metal complexes [17], and its excitation depends on the ligands, particularly when these last ones participate in a strong metal-ligand π-bonding, such a charge transfer is possible.

Moreover, these transitions can produce an intermolecular redox reaction when the excited states are promoted from the metal d orbitals to the orbitals (π, π*, σ and σ*) of the ligand. Many papers have been published regarding the photochemistry of NaNP, and only a handful have been made about other nitroprusside salts. Transition metal nitruprussides, form a family of coordination polymers with demonstrated potential applications for small molecules separation and storage, battery materials, sensors and actuators, information storage, photo-catalysis, in healthcare and other areas [14].

Nevertheless, their stability in the presence of water molecules and under the light incidence, two usually coinciding conditions in a normal environment, remains poorly documented and only recently has been studied [18,19]. This contribution discusses the accumulated information by our research group on the photochemistry of that family of materials, in conditions close to that expected during their applications. The light-induced electronic transitions Fe $\rightarrow$ π*NO modifies the electronic structure of metal nitroprussides and their physical properties. This has the potential for applications through materials with light-driven functional properties. This subject is also considered in this feature paper.

## 2. Photochemistry of the Nitroprusside Block

Although the metal nitroprussides were synthesized in 1848, the crystal structure of the sodium nitroprusside was determined a few decades later [2]. From this last study, we now know that the NP ion has approximately $C_{4v}$ symmetry, being all the C-N distances similar to other cyanide complexes and that the N-O distance is similar to the one found in NOX, where X is a halogen. However, the Fe-NO distance is short enough to suggest a triple bond between the Fe and N atoms, where the two π-bonds lay on the antibonding π*(NO) and the σ-bond goes to an empty d-orbital from the iron atom [20]. The work of Manoharan et al., using SCCC-MO calculations, gave valuable information about the ordering of the energy levels on the nitroprusside ion [21]. The resulting energy is shown in Figure 1. The highest-filled molecular orbitals, HOMO, in the ground state (GS) of the NP ion are represented as $(6e)^2(2b_2)^2$, where the former has $d_{xz}$ and $d_{yz}$ character but also contains about a quarter of π*(NO) character, and the second one has around 85% $d_{xy}$ character, with the remaining 14% being π(CN) and 2% π*(CN) contributions.

On the other hand, the lowest-unfilled molecular orbital, LUMO, is labeled as 7e and is around 73% of π*(NO) character, completed with small contributions from the metal, $d_{xy}$, $d_{xz}$, $d_{yz}$ and contributions from the σ(CN), π(CN) and π*(CN) orbitals. Because of the π*(NO) has a high contribution to the HOMO 6e, which has mainly metal character, a π-back bonding interaction between the Fe atom and the NO moiety can be inferred. This leads to a significant decrease in the electron density on the iron atom, reducing its π-back bonding interaction with the CN ligands, and in consequence, the electron density found at their 5σ orbital lowers.

This remains well documented from XPS spectra of metal nitroprussides [22]. The large contribution of the π*(NO) orbital to the 7e molecular orbital is probed by the frequency for the ν(NO) stretching band in the IR spectrum, which decreases when the $2b_2$ ($d_{xy}$) $\rightarrow$ 7e (π*NO) charge transfer increases [20]. For sodium nitroprusside, this transition is observed at 498 nm, which explains the red-purple color of crystals of that sodium salt. For transition metal nitroprussides, crystals of a varied diversity of colors (and absorption bands) are formed related to the combination of light absorption via MLCT and d-d transitions for the metal linked at the N end of the CN ligands [23]. Such d-d transitions have a minor impact on the photochemistry of metal nitroprussides.

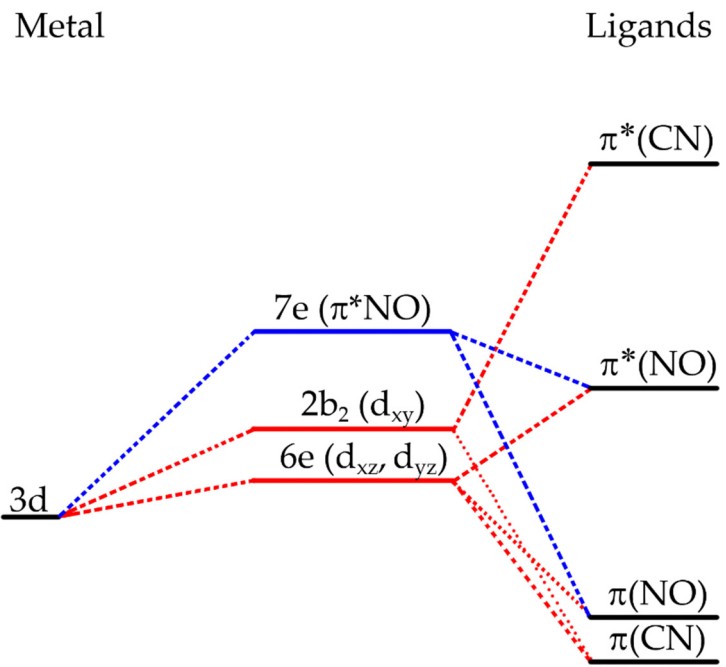

**Figure 1.** Ordering of the MO of NP, as proposed by Manoharan et al. [21]. The LUMO is represented in blue and the HOMO in red.

In 1977, Hauser et al. [24,25] observed a light-induced metastable state recording Mössbauer spectra at low temperature while a single crystal of NaNP was irradiated with a blue-green laser beam. With this technique, these authors found a spectrum that was different from the one obtained for the ground state (GS) measured without the laser beam irradiation. The spectrum collected during the irradiation was only visible at a certain angle.

Moreover, the irradiation must be continuous to produce this new state, so they ruled out the possibility of the new state being generated by light absorption or inelastic scattering. After these pioneer works, Zöllner et al. [26] studied by DSC (differential scanning calorimetry) measurements the thermal behavior of an irradiated sample of NaNP and discovered two long-lived metastable states (MS1 and MS2), the second decaying at a lower temperature; also, they found that the production of the MS states is exhibited in both single crystals or powders and even in other nitroprusside salts.

Subsequent experimental and theoretical studies [27–31] have enlighted a consistent picture on the subject: the light-induced metastable states are linkage isomers of the nitroprusside block, one of which has the O end of the NO bonded to the iron atom, the isonitrosyl $\eta^1$-O (MS1), and the other one has the NO side-bounded, $\eta^2$-NO (MS2) [32] (Figure 2).

These two metastable states result from the irradiation with a light source in the range of $\lambda$ = 350 to 580 nm while the sample is maintained underneath 100 K [33]; however, when the temperature is raised to 165 K, these MS decay to the GS again, and thus it is evident that the population of these metastable states depends on the polarization direction and wavelength of the light beam used [34]. Furthermore, these two states can be converted into one another by irradiating the sample with light of appropriate energy, 620–850 nm (Figure 2). The metastable states can also be de-excited by raising the temperature or irradiating the sample [26].

Further studies, recording SQUID (superconducting quantum interference device) magnetic data for nickel nitroprusside, $Ni[Fe(CN)_5NO]\cdot xH_2O$, at 5 K, irradiating the sample with light from an $Ar^+$ laser, $\lambda$ = 475 nm, revealed the appearance of magnetic order at low temperature. Such cooperative magnetic interaction disappears when the sample is warmed above 200 K [35,36]. Such behavior was explained as due to the light-driven

electron transfer from the iron atom to the NO ligand, resulting in the appearance of two new paramagnetic centers in the solid, on the iron atom and the NO ligand.

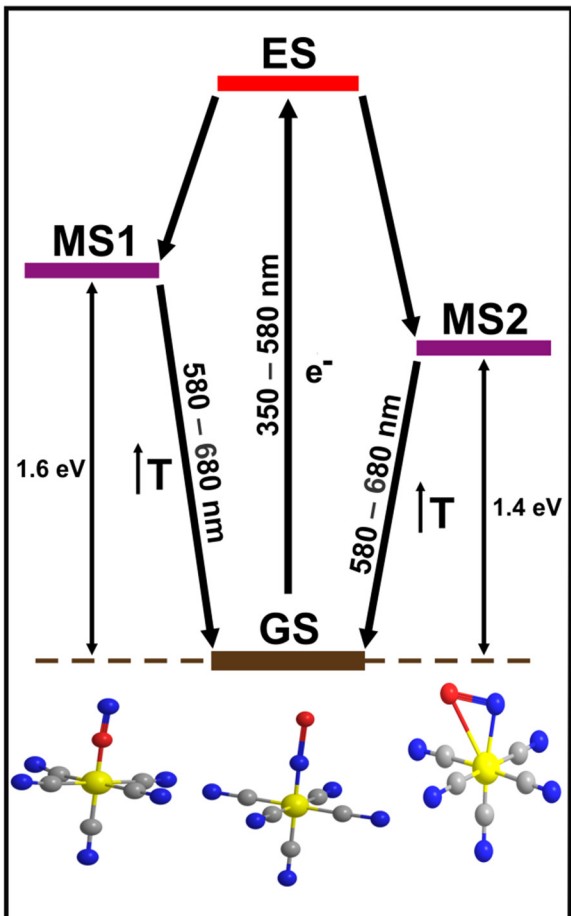

**Figure 2.** Excitation of isomeric transitions for the NO⁺ ligand by absorption in the UV–vis spectral region.

The observed cooperative magnetic order results from their superexchange interaction with the two unpaired electrons on the Ni atom. The authors reported the appearance of a new IR band at 1821 cm$^{-1}$, close to the frequency (1831 cm$^{-1}$) corresponding to the MS1 of sodium nitroprusside [37]. This suggests that the photo-induced magnetic order in nickel nitroprusside is related to a light excited state (ES), which then decays to the MS1 and MS2 states for the NO group (Figure 2). Such interpretation is congruent with the corresponding Mössbauer spectra recorded at low temperature (300 mK), which identified the presence of Fe(III) species in the irradiated sample [8]. This provides a conclusive clue to understanding the photochemistry of metal nitroprussides and their relationship with the MLCT in this ion.

From the nature of the HOMO and LUMO orbitals described above, these metastable states are considered as a d-π*(NO) one-electron excitation, however, Güdel proposed that the nature of the MS cannot be due to one electron configuration, so the MS can also be produced by a d-d transition added to the dominant MLCT [33]. These states have also been found in other nitrosyl complexes with transition metals when irradiated within the 350–580 nm wavelength range, either in 3D or 2D structures [26,38].

### 3. Photodecomposition: A Consequence of the Photoinduced Metal-Ligand-Charge-Transfer

#### 3.1. Photodegradation Studies on Sodium Nitroprusside (NaNP)

The photolysis of NaNP aqueous solution has been the subject of intense studies due to its importance in medical issues as a potent hypotensive agent. In 1963 Mitra et al. [39] reported that a solution of NaNP is acidified when irradiated with an unfiltered mercury discharge lamp; this effect was reversible provided the irradiation was for a short time; however, under prolonged irradiation, irreversible changes and secondary reactions appeared to take place. The authors proposed that the changes in the NaNP solution were triggered by the photolysis of the NP block, which releases the NO ligand as $NO^+$. Subsequent detailed studies covering spectroscopic and analytic techniques, mostly those from the groups of Wolfe, Stockel and de Oliveira [40–42], agree with a general pathway presented in Scheme 1.

**For T = $Na^+$, $Mn^{2+}$, $Zn^{2+}$, $Cd^{2+}$:**

$$T[Fe^{II}(CN)_5NO] \rightleftharpoons T^{2+} + [Fe^{II}(CN)_5NO]^{2-} \tag{1}$$

$$[Fe^{II}(CN)_5NO]^{2-} \xrightarrow[H_2O]{h\nu} \left[^-NC-Fe^{III}_\vdots NO\right]^{\ddagger} \longrightarrow [Fe^{III}(CN)_5H_2O]^{2-} + NO^\bullet \tag{2}$$

$$[Fe^{III}(CN)_5H_2O]^{2-} + NO^\bullet + 3\,H_2O \longrightarrow [Fe^{II}(CN)_5H_2O]^{3-} + NO_2^- + 2\,H_3O^+ \tag{3}$$

**For T = $Co^{2+}$, $Ni^{2+}$, $Cu^{2+}$:**

$$T[Fe^{II}(CN)_5NO] \xrightarrow[H_2O]{h\nu} \left[^{\delta-}T{\leftarrow}NC{\leftarrow}Fe^{III}\cdots{\overset{\delta+}{N}}O\right]^{\ddagger} \longrightarrow T[Fe^{II}(CN)_5NO] + H_2O \tag{4}$$

**For T = $Fe^{2+}$:**

$$Fe[Fe^{II}(CN)_5NO] \xrightarrow[H_2O]{h\nu} \left[Fe^{2+}{-}NC{-}Fe^{III}_\vdots NO\right]^{\ddagger} \longrightarrow Fe[Fe^{II}(CN)_5H_2O] + NO^\bullet \tag{5}$$

**Scheme 1.** General photodegradation pathways for transition metal nitroprussides as a function of the external metal (T).

The photodegradation of the nitroprusside ion in solution begins with the formation of the ES due to the intra-molecular electronic transition 6e → 7e resulting from the photon absorption in the blue region (c.a. 400 nm). As we noted before, orbital 6e has a marked bonding character reflecting the π-back bonding interaction of the Fe(II)-NO bond, whereas orbital 7e has a predominant π*(NO) character; hence, after the 6e → 7e transition, the resulting Fe(III)-NO bond is significantly weakened respect to the ground state since now the interaction is only of σ nature.

Thus, such destabilizing effect can give rise to the heterolytic rupture of the Fe-NO bond, leading to the release of radical NO and the formation of $[Fe^{III}(CN)_5H_2O]^{2-}$ ion after the incorporation of a water molecule to complete the coordination sphere of the central Fe(III) atom (step **2** in Scheme 1).

The effect of the solvent appears to be important in assisting the photo-degradation process since experiments performed in neat conditions [18] prevent the NO release, likely due to the cage effect. The NO release has been documented not only for nitroprusside aqueous solutions but also for nitroprusside dispersed in several solvents [41,43].

The formation of $[Fe^{III}(CN)_5H_2O]^{2-}$ ion and the release of NO to the solution are followed by several secondary photo- and redox-reactions that account for the observed acidification of the solutions and also yield as byproducts new species like $[Fe^{II}(CN)_5H_2O]^{3-}$, $NO_2^-$, $NO_3^-$, $CN^-$ and Prussian blue (PB). The extent of these reactions depends on the experimental conditions during the photodegradation test, as are the intensity and duration of the irradiation, sample concentration, pH, and level of oxygen dissolved. The $[Fe^{II}(CN)_5H_2O]^{3-}$ species can be formed by either thermal- and photo-reduction of the primary $[Fe^{III}(CN)_5H_2O]^{2-}$ ion [41,42,44], or by a redox reaction between this last species and the released NO molecules, leading to $NO^+$ cation that forms promptly $NO_2^-$ (step **3** in Scheme 1). In addition, NO can also be consumed in aerobic oxidation with the $O_2$ dissolved in the solution to form $NO_2^-$ and $NO_3^-$ anions [45].

The key role of the $6e \rightarrow 7e$ electronic transition is highlighted when studies are performed using monochromatic radiation [40,46]; hence, several works have reported that irradiating the nitroprusside solutions at the wavelength corresponding to the $2b_2 \rightarrow 7e$ transition (near 500 nm), no release of the nitrosyl radical is detected. The explanation lies in the fact that the $2b_2$ orbital is almost entirely composed of the iron $d_{xy}$ orbital, i.e., it has a marked nonbonding character; therefore, the promotion of an electron to the $\pi^*(NO)$ orbital does not influence significantly the strength of the Fe-NO bond.

An interesting question relies on whether the NO release competes with the linkage metastable isomers MS1 and MS2, or it happens from the metastable states as a two-step consecutive mechanism; this is a difficult task because photodegradation is usually observed after prolonged irradiation times at room temperature and in solution phase, i.e., under experimental conditions for which MS1 and MS2 decay rapidly and cannot be detected with steady-state techniques.

Up to now, there has been no consensus regarding this issue. Several works have postulated the consecutive mechanism [46–48] on the theoretical grounds that the structure of MS1 and MS2 could favor the Fe-NO bond rupture [49,50]. On the other hand, the work of Lynch et al. [37] supports the competitive mechanism for NaNP solutions at room temperature based on the picosecond transient Infrared Spectroscopy technique, with which they were able to trace simultaneously the appearance of MS1, MS2 and NO species.

*3.2. Effect of T in the photodegradation of T[Fe(CN)$_5$NO]·xH$_2$O*

The extent of the nitroprusside photolysis depends on the efficiency of the electronic reorganization within the ES; conditions that favor a strong Fe(III)-NO σ interaction provoke a fast deactivation of the ES, preventing bond rupture; on the contrary, as the σ interaction becomes weaker, the life-time of the ES is expected to be longer, thus, increasing the probabilities for Fe-NO bond rupture [46]. As can be inferred, there is room for controlling the photolysis of nitroprusside-containing materials by tuning the electronic properties of the internal block.

Based on this hypothesis, our research group tested the photo-stability of a series of aqueous 3D transition metal nitroprusside dispersions in order to study the influence of the external cation (T) bonded to the N ends of the T[Fe(CN)$_5$NO]·xH$_2$O structure, where T = Mn, Fe, Co, Ni, Cu, Zn and Cd [18]. For this endeavor, we subjected the aqueous dispersions to an intense white light lamp and analyzed the resulting products by FT-IR and Mössbauer Spectroscopies along with potentiometric measurements to track the pH variations during the photoreactions.

Since the ligands in the nitroprusside block are active in infrared absorption, giving well-localized narrow and intense bands, FTIR is a valuable technique to sense the changes involving them; at the same time, small variations in the coordination of the internal Fe atom can be probed with Mössbauer spectroscopy. Thus, these techniques allow for tracing

the loss of the NO ligand from the nitroprusside block and a facile identification of the degradation products $[Fe^{III}(CN)_5H_2O]^{2-}$ and $[Fe^{II}(CN)_5H_2O]^{3-}$.

Our results evidenced two limiting situations. On one hand, when T = Mn and Zn, the 3D compounds completely dissolve in water, and the recovered materials (by rotoevaporating the solutions) show the typical spectroscopic signs encountered for NaNP photodegradation, including a significant decrease of the solution pH. On the other hand, for T = Co, Ni and Cu, the compounds are practically insoluble, which permits the sample recovery by centrifugation; in these cases, the pH does not vary throughout the reaction, and the recovered solids do not display spectroscopic variations respect to the original materials.

The case of the Cd compound presents an intermediate behavior: the insoluble fraction remains unaltered and the soluble fraction is photodegraded. From these results, it is possible to rationalize the influence of T on the photodegradation tendency: 3D nitroprusside of metal cations with low polarizing power are easily ionizable in water, which liberates the internal building block and thus increases the electronic density in the N end of the CN ligands (step **1** in Scheme 1); this charge is then available for the Fe(III) atom to stabilize the photo-excited state, allowing longer a life-time that favors the rupture of the Fe(III)-NO bond (step **2** in Scheme 1).

In contrast, in the nitroprussides where T forms stronger T-NC bonds due to the significant polarization of the electronic density in the CN 5σ orbital toward the cation, the available electronic density in the nitroprusside block is lower; as a consequence, the σ interaction Fe(III) ← NO in the ES becomes stronger and leads to rapid quenching of the ES, frustrating the release of the nitrosyl radical from the nitroprusside block (step **4** in Scheme 1).

The case of the FeNP is quite interesting because contrary to the expected behavior owing to its polarizing power (intermediate between Mn and Co) not only the soluble but also the insoluble fraction presents spectroscopic footprints of degradation, which are now limited to the $[Fe^{II}(CN)_5H_2O]^{3-}$ species only. In addition, a careful analysis of the Mossbauer spectrum at 5 K suggests the presence of $Fe^{3+}$ cation in a high spin configuration that is external to the building block.

All the experimental evidence points to the formation of the mixed-valence compound $Fe[Fe^{II}(CN)_5H_2O]$, similar to PB. Inspired by this finding, we performed a series of experiments where several 3D TNP powder samples were extensively irradiated, with FeNP as the only compound that showed traces of photodegradation, expressed as the occurrence of the mixed-valence compound.

This result not only indicates the assisting role of the water in the NO release from the nitroprusside block but also demonstrates the unusual sensitivity of the FeNP toward photodegradation, which is clearly related with the PB-like species. In the initial ES, the configuration $Fe^{2+}$–NC–Fe(III)–NO leads to the more stable $Fe^{3+}$–NC–Fe(II)–NO by a metal-to-metal electron transfer from the external HS $Fe^{2+}$ to the internal LS Fe(III) mediated by the CN ligand, leading to a more stable configuration that significantly stabilizes the ES (step **5** in Scheme 1).

### 3.3. Effect of Pyridine (Py) in the Photodegradation of T(Py)₂[Fe(CN)₅NO]

To obtain further insights on the influence of the nitroprusside electronic structure on the material photostability, we prepared and evaluated a series of 2D transition metal nitroprusside derivatives containing pyridine (Py) as pillar molecules: $T(Py)_2[Fe(CN)_5NO]$ [19]. Effectively, the treatment of the 3D materials with an excess of the organic molecule drives the inclusion of it in the coordination sphere of the external metal as axial ligands, which liberates the axial cyanide ligand of the nitroprusside block and transforms the initial hydrated 3D structure into an anhydrous 2D structure [11].

As in the previous study, we tested the photostability of aqueous dispersions containing these 2D compounds as a function of the external metal T. For the sake of comparison, experiments in dark conditions were first performed. The most striking feature was the partial or complete recovery of the 3D phase due to the loss of the Py molecules from the

metal coordination sphere; the 3D phase was detected by FTIR, and the free Py molecules can be sensed due to the basification of the dispersed medium.

The 2D → 3D transformation is more prominent as the polarizing power of T decreases; for instance, while for the Ni compound the 3D phase is barely detected, for the Mn and Zn counterparts, the whole samples are recovered in the soluble fraction as the 3D phases. For the rest of the samples, both phases coexist with a 2D/3D composition that is consistent with the cation polarizing power order. Using the Cu compound as a model, we were able to tune the extent of Py losses by varying the ionic force of the medium, since high salt concentration prevents the substitution of Py by water molecules in the coordination sphere of T.

With this in mind, we then performed the analogous experiments under white light illumination. The recorded degradation effects for the case of metals with low polarizing power (Mn, Zn and Fe) were similar to those encountered during the photostability study of the 3D TNP. This was expected because of the fast 3D formation at the initial stages of the experiment; the pH behavior is consistent with this notion: it increases suddenly due to Py release from the nitroprusside structure and then begins to decrease as a consequence of the photolysis reactions.

The analysis of the data corresponding to the compounds containing cations of higher polarizing power demonstrated that, contrary to the 3D TNP materials, a certain level of photodegradation can be detected together with the ubiquitous 3D phase. In order to rationalize this finding, we analyzed the NO release as a function of the ionic force of the medium for the Cu compound. The obtained results indicated that NO losses are more extensive as the ionic force becomes higher; at the same time, the same direct relation can be found between the ionic force and the extent of Py losses, which is the inverse behavior seen during the in-darkness experiments. Therefore, it seems that not only does NO and Py release go hand-in-hand under illuminating conditions but also, more importantly, NO release triggers the Py release.

The key factor for understanding the situation relies in the differential structural factor that appears in the nitroprusside block on 2D formation: the free axial cyanide ligand ($CN_{axial}$). For the cases where T is a low polarizing cation, the fast Py release provokes the sudden formation of the 3D phase, which becomes ionized, and this process liberates the nitroprusside block from the influence of T; therefore, the free $CN_{axial}$ initially encountered in the 2D structure does not have the opportunity to influence the photodegradation process. Now well, the picture is different when the 2D structure contains a metal with high polarizing power.

As it was explained, in this case, T retains the Py molecules in its coordination sphere, thus, retarding the 2D → 3D transformation; in other words, the $CN_{axial}$ of the nitroprusside block remains unbonded to metal centers. This allows the $CN_{axial}$ to play part in the electronic rearrangement that takes place after photoexcitation (step **1** in Scheme 2); in fact, since now the available electronic density within the nitroprusside block is higher with respect to the case when all the CN ligands are forming T-NC bonds, the extra stabilization of the Fe(III) atom in the ES favors the rupture of the Fe-NO bond (step **2** in Scheme 2).

That explains why the 2D materials are more sensitive to photodegradation that the 3D counterparts when T = Co, Cu and Ni. This proposed pathway is consistent with the direct proportionality between the ionic force of the medium and the extent of NO release: as the ionic force increases, the dielectric constant of the medium decreases, causing the excess electronic density around the free $CN_{axial}$ to be less dispersed throughout the medium, i.e., more available for stabilizing the photo-induced ES in the nitroprusside block.

Yet, what about the observed parallel behavior between NO and Py loses? Again, the free $CN_{axial}$ plays a fundamental role. The release of the NO ligand produces $[Fe^{III}(CN)_5H_2O]^{2-}$ residues throughout the structure; the extensive electronic rearrangement that takes place within the aquopentacyanide complex provokes the $CN_{axial}$ to bear higher electronic density with respect to the case of the nitroprusside complex. As a consequence, the $CN_{axial}$ ligand

becomes more nucleophilic and could displace one Py molecule by attacking the metal center of an adjacent layer, thus, forming a local 3D substructure (step **3** in Scheme 2).

**Scheme 2.** Photodegradation pathways for the Co, Ni and CuPyNP samples.

Hence, it follows that, as the photolysis events are more frequent, the release of Py molecules from the structure becomes more extensive. Clearly, the extent of the pillar's loss depends on its nucleophilicity. Thus, when Py is substituted by a more nucleophilic derivative, such as 4-methoxypyridine [51], it is apparent that the rate of NO loss increases respect to the rate of molecule loss.

Altogether, several useful conclusions can be drawn about the potential of transition metal nitroprussides to tune the photodegradation response, which are summarized in Table 1 for the studied compounds. For instance, the series of 3D TNP materials demonstrate that there is a clear gap in the photo sensibility as a function of the metal nature. However, the introduction of pillared molecules appears to open up possibilities for finer tuning; in this regard, the variation of other structural properties (the nucleophilicity of the organic ligand) or the medium conditions (the dielectric constant) could also be helpful. Controlling the NO release in photo-sensible systems is crucial in the design of functional materials with biomedical applications [52,53].

**Table 1.** The photostability of the aqueous dispersions for the studied 3D and 2D transition metal nitroprussides *.

| T | T[Fe(CN)$_5$NO]·xH$_2$O (3D) | T(Py)$_2$[Fe(CN)$_5$NO] (2D) |
|---|---|---|
| Mn | Low | Low |
| Fe | Low | Low |
| Co | High | Medium |
| Ni | High | High |
| Cu | High | Medium |
| Zn | Low | Low |
| Cd | Medium | - |

* Experimental conditions: illuminating source: white-light lamp ($8 \times 10^4$ lux); sample concentration: 0.5 g/L; reaction volume: 100 mL; reaction time: 4 h. See other details in references [18,19].

## 4. Role of the Photo-Induced Charge Transfer in the Physical and Functional Properties

The relatively high stability (lifetime $\tau > 10^7$ s below 160 K) for the photo-induced metastable states (MS1 and MS2) together with the GS state (Figure 2) represent three available possible light-driven configurations in the nitroprusside ion to modify the physical and functional properties of materials based on that building block. The existence of such long-lifetime meta-stable states stimulated the study of crystals of sodium, barium and guanidinium nitroprussides as materials for holographic information storage in the past [54].

In addition, the orientation of the Fe-NO group can be tuned through a controlled $2b_2$ (xy) $\rightarrow$ 7e ($\pi$*NO) charge transfer within the GS, enlarging such possibilities for nitroprusside-based materials. That last effect is observed, for instance, during the 3D to 2D structural transition, which is probed as a progressive decrease in the frequency of the $\nu$(NO) stretching vibration but without reaching the frequency values corresponding to the meta-stable states [55].

The above-discussed photo-induced magnetic order in nickel nitroprusside is a typical example of light-driven physical properties in metal nitroprusside taking advantage of the MLCT by photon absorption in a certain energy region [35,36]. Such a charge transfer leads to the appearance of an unpaired electron on the iron atom and, at the same time, to a reduction in the electron density on this atom.

This last effect reduces the charge density accumulated at the CN 5$\sigma$ orbital; however, the Ni(2+) has a sufficiently high polarizing power to participate in a strong superexchange magnetic interaction with the iron atom, thereby, resulting in the reported photo-induced magnetic order at low temperature. The unpaired electrons in the Ni and iron atoms are found in orthogonal orbitals, $e_g$ and $t_{2g}$, respectively, and the magnetic interaction between them has a ferrimagnetic character (Figure 3A). This is a typical example of a cooperative physical property induced by the sample illumination.

For the Mo analog, $Cs_{1.1}Fe_{0.95}[Mo(CN)_5NO]\cdot4H_2O$, the photo-switchable ion conduction was documented [56]. In the crystal structure of that material, six molecular blocks ($Mo(CN)_5NO$) are arranged in such a way that their six unbridged NO groups are oriented toward the center of a cube forming a cavity of ca. 5 Å. One of the water molecules is found coordinated to the iron atom, which behaves as a Lewis acid, inducing the release of a proton from the coordinated water molecule.

The remaining water molecules form a chain of hydrogen-bonding interactions where the coordinated water molecule and the O end of the NO group participate. Such a network of hydrogen-bonding interactions supports fast ionic conduction, $1 \times 10^{-3}$ Scm$^{-1}$, for the released proton, via its hopping between water molecules (Grotthuss mechanism), in the non-irradiated sample. When the sample is irradiated with a light of 532 nm, the ionic conductivity decreases to $6 \times 10^{-5}$ Scm$^{-1}$, which is accompanied by a decrease in the frequency of the $\nu$(NO) vibration.

Such changes were ascribed to a photo-isomerization of the NO ligand, breaking the chain of hydrogen-bonding interactions and, from this fact, the observed reduction in the ionic conductivity (Figure 3B); once the irradiation ceases, the conductivity progressively recovers the initial value [56]. More recently, the same research group has explored the use of a dysprosium-containing nitroprusside complex as a reversible photo-switchable nonlinear-optical crystal [57]. Upon irradiation at 473 nm, there is a notable increase in the second harmonic generation signal due to the large hyper-polarizability of the –ON+ group belonging to the linkage isomer MS1, which is depleted if the crystal is brought back to its ground state by irradiating at 804 nm.

Recently, the formation of a salt of the nitroprusside ion with polar organic cation, e.g., dimethylammonium ($Me_2NH_2^+$), was evaluated as a route to obtain light-tunable ferroelectric materials [58]. That organic cation has a permanent dipole moment of 2.2 Debye. Such hybrid solids are obtained by evaporation of an aqueous solution of sodium nitroprusside and the organic cation as hydrochloride. From the obtained solid, crystals

of $(Me_2NH_2)Na[Fe(CN)_5NO]$ are separated and then submitted to a variable-temperature structural study, in the 298 to 458 K range, complemented with DSC data.

This material undergoes a reversible structural transition that is easily detected in the recorded DSC curves as exothermic/endothermic peaks at ca. 423 and 408 K. These thermal effects and the XRD powder patterns provide a conclusive clue on the occurrence of an order-disorder phase transition. The structure for the phase below the transition temperature ($T_C$) corresponds to a polar space group *Pna*$2_1$, where the organic cation behaves as a charge balancing cation occupying the cavities in a framework formed by $Na[Fe(CN)_5NO]$ units, interacting with the N ends of the ligands through hydrogen bonds.

Once the material is heated above $T_C$, the structure changes to a centrosymmetric space group (*Pnam*) related to a certain rotational disorder for the organic cation. Such structural change has relevant implications for the material physical properties. Below $T_C$, it behaves as a ferroelectric material, and the observed structural change results in a first-order ferroelectric phase transition. Below Tc, all the organic cation dipole moments appear ordered (Figure 3C). According to the recorded IR spectra before and under irradiation with light of 532 nm, in this material, both MS1 and MS2 meta-stable states can be excited.

From this fact, the authors advanced that such a photo-induced effect opens the possibility to obtain multiple-states ferroelectric materials with attractive potential applications. An analog study was reported for $(Me_2NH_2)K[Fe(CN)_5NO]$ with the additional attractive that its authors report a uniaxial positive thermal expansion below $T_C$ but above the transition temperature, the three axes elongate without a notable anisotropy [59]. Considering the photo-isomerization in the nitroprusside building block, its effect on both the ferroelectric state and the atypical thermal expansion is anticipated but not documented.

Recently, 2D ferrous nitroprussides pillared with organic ligands have emerged as a series of hybrid inorganic-organic solids with thermally-induced spin-crossover (SCO) behavior [12,13,60]. Such an unusually large number of compositions with SCO was ascribed to the role of the NO group as an electron buffer to modulate the electron density found at CN $5\sigma$ orbital in the equatorial CN ligands [60]. This allows the tuning of the local crystal field sensed by the iron atom to have a 10Dq value appropriate to stabilize the low spin (LS) state ($e_g^0 t_{2g}^6$) on the sample cooling.

If the sample in the LS is irradiated with light of a wavelength appropriate to induce a Fe to NO charge transfer, the electron density in that molecular orbital ($5\sigma$) lowers, diminishing the value of 10Dq, and the spin state will be perturbed, favoring the LS to HS (high spin) transition. That effect has not been studied; however, it is expected. In that case, the sample irradiation would be a route to commute between the LS and HS states with consequent potential applications (Figure 3D).

In the interlayer region of 2D transition metal nitroprussides, the NO group is found as a dangling ligand with a negative charge density at its O end. This is an appropriate adsorption site for small quadrupolar molecules, e.g., $H_2$, $N_2$, $CO_2$ and $CS_2$. The presence of such guest molecules in that region can be probed as a certain frequency shift for the $\nu$(CN) stretching band accompanied by a change in its splitting [55]. These effects could be amplified through the sample illumination, within the GS for the NO ligand.

Figure 3 illustrates the main mechanisms involved in the light-driven functional properties related to the photo-induced NO isomerization discussed above. The study of molecular materials and those of hybrid inorganic-organic nature is an emerging research area, and many examples where the nitroprusside ion forms part of these materials could be reported in the near future. In such materials, the photo-induced isomerization of the NO ligand appears like a route for tuning their physical and functional properties.

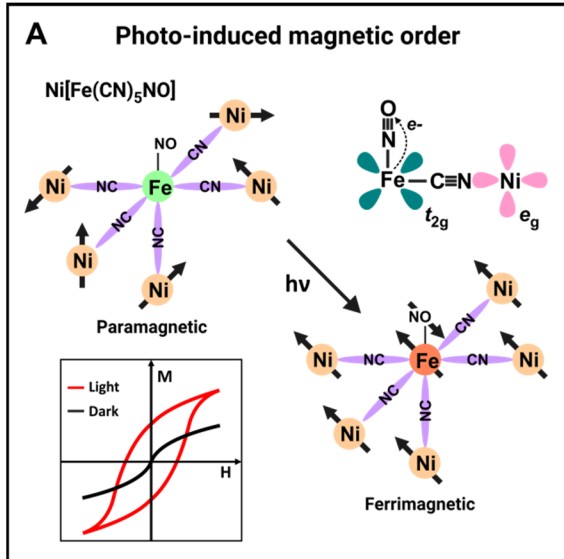

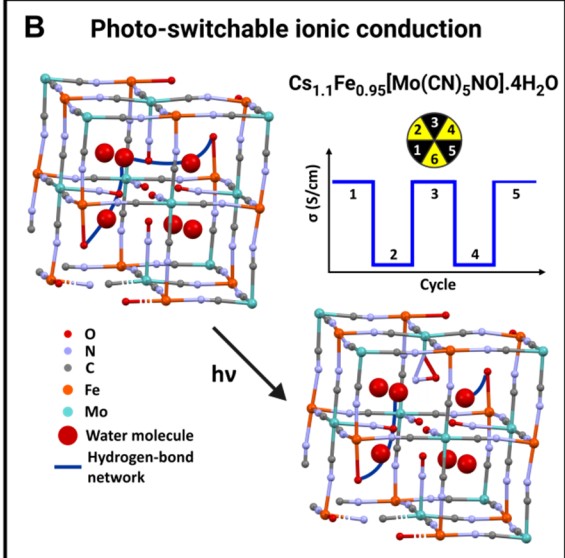

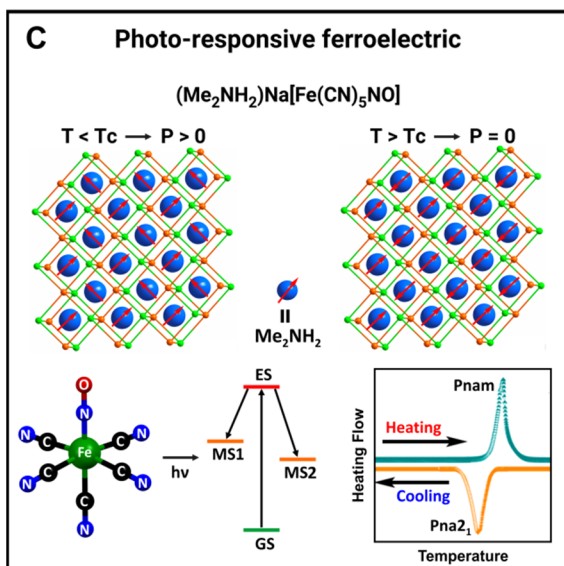

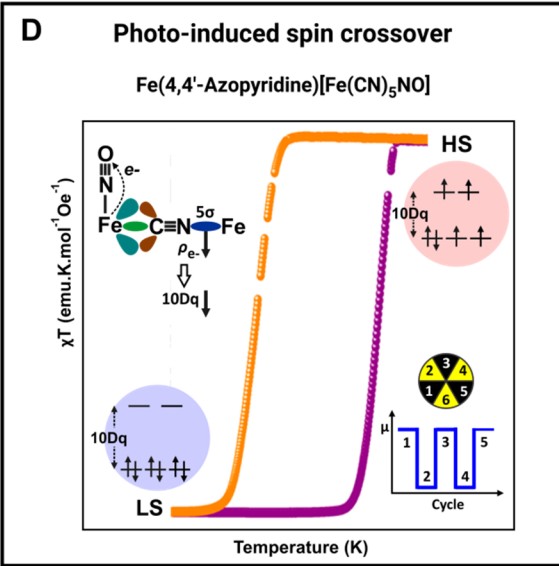

**Figure 3.** Photo-driven functional properties of materials based on the nitroprusside ion. Images prepared from the reported results in: (**A**) Ref. [36], (**B**) Ref. [56], (**C**) Ref. [58] and (**D**) Ref. [60].

## 5. Concluding Remarks

Nitroprussides exhibit interesting physical and functional properties—particularly the coordination polymers formed with transition metal ions. Such properties have been recently summarized and include their potential application for small molecule separation and storage, battery materials, sensors, actuators, information storage, optical materials, photo-catalysis, healthcare, etc. In the presence of water molecules, including those found in their framework as structural or absorbed water, we discussed the light absorption and induced photo-isomerization results in the liberation of the photo-active nitrosyl group with the consequent material degradation.

In this paper, the understanding of the degradation mechanisms in the presence of light and water received particular attention. In the absence of water molecules, the photo-isomerization of the NO group appears as a reversible process that can be used to modulate the physical and functional properties of metal nitroprussides. The photo-chemistry of the nitroprusside ion in other solvents remains poorly documented, and considering the

potential application of materials based on this building block, this is a subject that deserves to be considered in further studies. Molecular and hybrid inorganic–organic solids are opening new opportunities to prepare materials where different physical properties are combined to obtain novel functional materials.

This will be an active research area for metal nitroprussides in the future for applications in spintronic, photo-optical devices, sensors and actuators and smart materials with the ability to respond to external agents as observed in solids with a spin-crossover effect. In this contribution, some of these potentialities are briefly summarized, and others are anticipated. This feature paper was prepared with the intention of helping to understand the photochemistry of nitroprusside-based materials and how to control the favorable and unfavorable effects.

**Author Contributions:** P.M.C., O.F.O. and E.R. had a similar level of contribution to its preparation, including the design, writing, artwork preparation and revision. All authors have read and agreed to the published version of the manuscript.

**Funding:** This research received no external funding.

**Acknowledgments:** The authors thank LNCAE (Laboratorio NaciAlmacenamiento de Energía) for the access to its experimental facility. The preparation of this contribution was partially supported by the project SECITI/185/2021.

**Conflicts of Interest:** The authors have no conflict of interest. The information contained in this manuscript has a basic character and concerns new scientific knowledge. The data provided by the authors through this manuscript are available on request.

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
