# Peer review of "Photochemistry of Metal Nitroprussides: State-of-the-Art and Perspectives"

_2673-7256, doi:10.3390/photochem2020027_

Round 1
Reviewer 1 Report
This manuscript discusses the Photochemistry of Metal Nitroprussides, focusing on structural characteristics (2D to 3D), electronic properties and photodecomposition properties. The main interest is oriented on how to control their photodecomposition. Some of the authors have recently reviewed this type of materials (Transition metal nitroprussides: Crystal and electronic structure, and related properties, Coordination Chemistry Reviews, 2021, 434, 213764), as appears in reference 13 of the current manuscript. It is unclear to me what is the scope of the current manuscript. I see that recent efforts from the authors are appearing in this manuscript. In the current form of the manuscript, recent advances are “blended” with previous studies. In my opinion, if there is a justified reason for publishing a new review article, the recent advances should be discussed in a separate section, where the reader could clearly identify the progress and the contribution of the authors in this field.
I believe this manuscript is not adequate for publication in it’s current form.
Moreover, the authors should also consider the following points:
Lines 9-10: Authors mention that the photochemistry of metal nitroprussides is dominated by the electronic structure of the nitrosyl group. The metal, metal valency, metal-CN and metal-NO interactions are not affecting the photochemical properties? Please clarify.
Line 22: …”for tuning such properties”. Which properties? …“these opportunities”. Which opportunities? Please clarify.
Line 23: …“possible research subjects”. Not adequately described in the main discussion body and concluding remarks.
Line 34 (Introduction): Herein, “coordination polymers” are mentioned (the same also in line 422 of Concluding remarks). However, this term is not used throughout the main discussion body. Why is this happening?
Lines 91-96: the discussion should be restructured to be clearer.
Line 97: “DSC”. Please explain the term to the readers.
Line 113: “SQUID”. Please explain the term to the readers.
Line 144: “were motivated”. Please rephrase.
Line 163: “there can occurs”. Please rephrase.
The discussion related to the influence of the external cation (T) bonded to the N ends of the T[Fe(CN)5NO].xH20 structure, should be grouped under a separate section.
The use of pillars, i.e. Py, should be useful to be presented in a separate section.
Headings of sections 2 and 3 are the same. Please revise appropriately.
The development of 3D structures should be also described in a separate section.
A table summarizing pros/cons and compositions of 2D and 3D structures would be also useful for the reader.
Line 438: “This will be an active research area for metal nitroprussides in the future.” Why? Please elaborate on this (fundamental research, potential uses, applications etc).
Lines 443-444: Did the authors use supplementary materials? If not, please remove the lines.
Lines 445-453: Author contributions should be clearly described. Please clarify.
Lines 454-457: Please mention the funding sources, otherwise remove the lines.
Lines 458-462: Please respond to the data availability statement.
Author Response
Reviewer 1
General comment: This manuscript discusses the Photochemistry of Metal Nitroprussides, focusing on structural characteristics (2D to 3D), electronic properties, and photodecomposition properties. The main interest is oriented on how to control their photodecomposition. Some of the authors have recently reviewed this type of materials (Transition metal nitroprussides: Crystal and electronic structure, and related properties, Coordination Chemistry Reviews, 2021, 434, 213764), as appears in reference 13 of the current manuscript. It is unclear to me what is the scope of the current manuscript. I see that recent efforts from the authors are appearing in this manuscript. In the current form of the manuscript, recent advances are “blended” with previous studies. In my opinion, if there is a justified reason for publishing a new review article, the recent advances should be discussed in a separate section, where the reader could clearly identify the progress and the contribution of the authors in this field.
I believe this manuscript is not adequate for publication in its current form.
Response: That review paper concerns the crystal and electronic structures of transition metal nitroprussides, not their photochemistry. There is no blend between that review and this manuscript. This last one was prepared and submitted in response to an invitation by the journal of Photochemistry (MDPI). This manuscript is focused on the photoinduced charge transfer in the nitroprusside building block and its consequences on the stability of 3D and 2D transition metal nitroprussides, a subject recently documented by our research group. In such stability, the combined effect of light and the presence of water molecules plays a relevant role. Such photoinduced charge transfer is not only relevant for the stability of this family of materials in environmental conditions. This effect finds potential application in functional materials, among them, light-driven molecular magnets, proton conduction, multiferroic properties, and spin-crossover behavior. The light-induced Fe ®NO charge transfer removes charge density from the iron atom, weakening the ability of that last one to participate in a p-back bonding interaction with the CN ligands. This last effect leads to a reduction in the electron at the CN 5s orbital, and from this fact, it has the capability of modifying the crystal field splitting sensed by the metal linked a the N end of the CN ligands. This is a route for the light-driven of optical and magnetic properties in transition metal nitroprussides. This and many other photoinduced effects were not discussed in the mentioned review paper. We reiterate that there is no blend between that review and this manuscript.
Moreover, the authors should also consider the following points:
Comment 1: Lines 9-10: Authors mention that the photochemistry of metal nitroprussides is dominated by the electronic structure of the nitrosyl group. The metal, metal valency, metal-CN, and metal-NO interactions are not affecting the photochemical properties? Please clarify.
Response: Such effects appear discussed in the revised version. This manuscript discusses the role of the metal linked at the N end of the CN ligand on the stability of transition metal irradiated with withe light in the presence of water molecules, particularly related to the nature of the metal, its valence, and polarizing power. Of course, the metal-CN and Fe-NO interactions are relevant because they determine the metal-ligand-charge-transfer processes in this series of materials, which are closely related to their photochemistry and photo-degradation. These subjects are discussed in detail in the revised version.
Comment 2: Line 22: …”for tuning such properties”. Which properties? …“these opportunities”. Which opportunities? Please clarify.
Response: Such a statement appears in the manuscript abstract as general information about the subjects that are discussed in the manuscript, and these subjects are really discussed in the manuscript body, with emphasis considering this critical comment (from Reviewer 1). Yet, we modified this part in the following way: “…an opportunity for tuning their magnetic, electrical, and optical behavior, for instance”. In the next line, the phrase “these opportunities” refers to the “opportunity for tuning…” mentioned above.
Comment 3: Line 23: …“possible research subjects”. Not adequately described in the main discussion body and concluding remarks.
Response: Similar to the previous critical comment, that mention appears in the abstract as a statement of the manuscript content. Considering this critical comment, this subject appears discussed in detail in the manuscript-revised version.
Comment 4: Line 34 (Introduction): Herein, “coordination polymers” are mentioned (the same also in line 422 of Concluding remarks). However, this term is not used throughout the main discussion body. Why is this happening?
Response: The solids formed by the reaction of the nitroprusside anion with monovalent and divalent transition metals are usually named coordination polymers by the nature of their crystal structure. For the revised version, such a term is clarified.
Comment 5: Lines 91-96: the discussion should be restructured to be clearer.
Response: This suggestion was considered and that discussion was properly restructured. In the revised version, that segment appears as “In 1977, Hauser et al. [21, 22] observed a light-induced metastable state recording Mössbauer spectra at low temperature while a single crystal of NaNP was irradiated with a blue-green laser beam”.
Comment 6: Line 97: “DSC”. Please explain the term to the readers.
Response: In the revised version appears DSC (differential scanning calorimetry)
Comment 7: Line 113: “SQUID”. Please explain the term to the readers.
Response: In the revised version appears: “SQUID (superconducting quantum interference device).
Comment 8: Line 144: “were motivated”. Please rephrase.
Response: In the revised version that sentence was changed to “were triggered”.
Comment 9: Line 163: “there can occurs”. Please rephrase.
Response: The statement where that phrase is found was rephrased to “The formation of [FeIII(CN)5H2O]2- ion and the release of NO to the solution are followed by several secondary photo- and redox- reactions that account for the observed acidification of the solutions and also yield as byproducts new species like [FeII(CN)5H2O]3-, ..
Comment 10: The discussion related to the influence of the external cation (T) bonded to the N ends of the T[Fe(CN)5NO].xH20 structure, should be grouped under a separate section.
Response: We evaluated such interesting suggestion, which we believe is oriented to clarify the discussion. In that sense, we divided section 3 in three subsections: 3.1. Photodegradation studies on sodium nitroprusside (NaNP); 3.2. Effect of T in the photodegradation of T[Fe(CN)5NO].xH2O; 3.3. Effect of pyridine (Py) in the photodegradation of T(Py)2[Fe(CN)5NO].
Comment 11: The use of pillars, i.e. Py, should be useful to be presented in a separate section.
Response: Please, see the above response.
Comment 12: Headings of sections 2 and 3 are the same. Please revise appropriately.
Response: This was properly modified to “Photodecomposition: a consequence of the photoinduced metal-ligand-charge-transfer”
Comment 13: The development of 3D structures should be also described in a separate section.
Response: This manuscript is intended as a feature paper on the photochemistry of metal nitroprussides. The development of 3D structures appears discussed in the above-mentioned review.
Comment 14: A table summarizing pros/cons and compositions of 2D and 3D structures would be also useful for the reader.
Response: Thanks for the suggestion. We have added in the revised version of the manuscript a table (Table 1) summarizing the qualitative behavior of the photostability of the aqueous dispersions for the studied 3D and 2D transition metal nitroprussides, which we consider will be of general utility for the readers.
Comment 15: Line 438: “This will be an active research area for metal nitroprussides in the future.” Why? Please elaborate on this (fundamental research, potential uses, applications etc).
Response: In the revised version this was clarified in terms of: “This will be an active research area for metal nitroprussides in the future for applications in spintronic, photo-optical devices, sensors and actuators, and smart materials with the ability to respond to external agents, as it is observed in solids with spin-crossover effect”.
Comment 16: Lines 443-444: Did the authors use supplementary materials? If not, please remove the lines.
Response: This manuscript does not include Supplementary Materials
Comment 17: Lines 445-453: Author contributions should be clearly described. Please clarify.
Response: This appears clarified in the revised version: All the authors of this manuscript have a similar level of contribution to its preparation, which includes design, writing, artwork preparation, and revision.
Comment 18: Lines 454-457: Please mention the funding sources, otherwise remove the lines.
Response: This is a journal requirement, and must be included.
Comment 19: Lines 458-462: Please respond to the data availability statement.
Response: The following sentence was added “The data provided by the authors through this manuscript are available on request”.
Reviewer 2 Report
The manuscript of P. M. Crespo, O. F. Odio, and E. Reguera is a mini-review on the photochemistry of metal nitroprussides. The authors consider the photolysis of such materials in water and classify the dependence of photodegradation pathways on the nature of the metal. Then they analyze the effect of photoexcitation on the physical properties of mixed metal and inorganic-organic nitroprussides. The manuscript is clearly structured and easy to read.
The strengths of the manuscript are a clear explanation of the topics covered, the inclusion of recent publications in the review, and the presentation of the main results in pictures. The manuscript is clearly structured and easy to read.
Minor: Please correct the title of Part 3 according to its content.
Author Response
Response to the Reviewers´ critical comments
The authors have considered all the reviewers´ critical comments and suggestions to improve the manuscript quality and scope. Below we provide the response to these comments.
Reviewer 2:
The manuscript of P. M. Crespo, O. F. Odio, and E. Reguera is a mini-review on the photochemistry of metal nitroprussides. The authors consider the photolysis of such materials in water and classify the dependence of photodegradation pathways on the nature of the metal. Then they analyze the effect of photoexcitation on the physical properties of mixed metal and inorganic-organic nitroprussides. The manuscript is clearly structured and easy to read.
The strengths of the manuscript are a clear explanation of the topics covered, the inclusion of recent publications in the review, and the presentation of the main results in pictures. The manuscript is clearly structured and easy to read.
Minor: Please correct the title of Part 3 according to its content.
Response: Thank you for your suggestion. The title was modified to “Photodecomposition: a consequence of the photoinduced metal-ligand-charge-transfer”
Reviewer 3 Report
The authors comprehensively summarize the current knowledge about iron(II) nitroprusside. In the review paper, after explaining the electronic structure of nitroprusside, they discuss implications of photo-induced transitions to the stability of coordination compounds based on the nitroprusside ion in the presence of water molecules. Additionally, it is introduced that photo-induced electronic transitions of nitroprusside modify the physical properties and functionalities of the base materials referring some recent papers. The description of the electronic structure of nitroprusside, the photochemical properties in the presence of water molecules, and the photofunctionalities in the solid state of the nitroprusside-implemented materials will attract interest in the field of photochemistry. Thus, the manuscript will be suitable for publication in Photochem after several revisions as follows.
(1) The authors should refer to a few articles in this manuscript, a historical review of metal nitrosyl complexes (Coord. Chem. Rev. 1974, 13, 339) showing nitroprussides in a broader context, an article on photochromic effects in Mn (III) and Mn (II) Schiff-base and pentacyanonitrosylferrate (CrystEngComm, 2015, 17, 3866), and an article describing the photoswitching of nonlinear optics effects (Inorg. Chem. 2021, 60, 4, 2097).
(2) This manuscript is a “Review” paper, not “Article”. Please check the types of publications.
(3) The subtitle of 2 and 3 are the same. The authors should change them to suitable subtitles.
(4) Page 1, Line 29: There is a typo in “pentacyanonitrosylferrate(II)”.
(5) Page 5, Line 166: There are typos for “nitrite” and “nitrate”. The authors should check the font style.
Author Response
Response to the Reviewers´ critical comments
The authors have considered all the reviewers´ critical comments and suggestions to improve the manuscript quality and scope. Below we provide the response to these comments.
Reviewer 3:
The authors comprehensively summarize the current knowledge about iron(II) nitroprusside. In the review paper, after explaining the electronic structure of nitroprusside, they discuss implications of photo-induced transitions to the stability of coordination compounds based on the nitroprusside ion in the presence of water molecules. Additionally, it is introduced that photo-induced electronic transitions of nitroprusside modify the physical properties and functionalities of the base materials referring some recent papers. The description of the electronic structure of nitroprusside, the photochemical properties in the presence of water molecules, and the photofunctionalities in the solid state of the nitroprusside-implemented materials will attract interest in the field of photochemistry. Thus, the manuscript will be suitable for publication in Photochem after several revisions as follows.
Comment 1: (1) The authors should refer to a few articles in this manuscript, a historical review of metal nitrosyl complexes (Coord. Chem. Rev. 1974, 13, 339) showing nitroprussides in a broader context, an article on photochromic effects in Mn (III) and Mn (II) Schiff-base and pentacyanonitrosylferrate (CrystEngComm, 2015, 17, 3866), and an article describing the photoswitching of nonlinear optics effects (Inorg. Chem. 2021, 60, 4, 2097).
Response: These suggested references were considered and cited in the revised version, which are now references [3], [38], and [57].
Comment 2: (2) This manuscript is a “Review” paper, not “Article”. Please check the types of publications.
Response: This is an invited contribution from the journal, as a feature paper.
Comment 3: (3) The subtitle of 2 and 3 are the same. The authors should change them to suitable subtitles.
Response: In the revised version, Section 3 appears under the subtitle: “Photodecomposition: a consequence of the photoinduced metal-ligand-charge-transfer”
Comment 4: (4) Page 1, Line 29: There is a typo in “pentacyanonitrosylferrate(II)”.
Response: This was corrected for the revised version
Comment 5: (5) Page 5, Line 166: There are typos for “nitrite” and “nitrate”. The authors should check the font style.
Response: This was corrected for the revised version. In the revised version appears: NO2-, NO3-,
Round 2
Reviewer 3 Report
The manuscript has been properly revised along the reviewer’s comments, and become clear for readers. Thus, the present manuscript is suitable for the publication in PhotoChem.
Author Response
The authors thank the useful critical comments and suggestions from this reviewers